# Reducing Storage Losses of Organic Apples by Plasma Processed Air (PPA)

Robert Wagner [1,*], Thomas Weihe [1,*], Hauke Winter [2], Christoph Weit [1], Jörg Ehlbeck [1] and Uta Schnabel [1]

1   Leibniz Institute for Plasma Science and Technology, 17489 Greifswald, Germany;
    christoph.weit@freenet.de (C.W.); ehlbeckl@inp-greifswald.de (J.E.); uta.schnabel@inp-greifswald.de (U.S.)
2   Institute of Microbiology, University of Greifswald, 17489 Greifswald, Germany;
    hauke.winter@stud.uni-greifswald.de
*   Correspondence: robert.wagner@inp-greifswald.de (R.W.); thomas.weihe@inp-greifswald.de (T.W.);
    Tel.: +49-3834-554-3868 (R.W.); +49-3834-554-3848 (T.W.)

**Abstract:** The consumer demand for organic food including apples is increasing worldwide. Despite favorable environmental and health benefits, organic farming bears also disadvantages like high amounts of fruit losses due to storage rot. A novel treatment with plasma-processed air (PPA) to sanitize organic apples is investigated. The plasma source for the generation of PPA was operated at a frequency of 2.45 GHz, a power output of 1.1 kW and a gas flow of 18 standard liters per minute. The antimicrobial efficiency of the PPA was tested on the natural load of organic apples (cultivar Natyra) with a load ranging from $10^4$ to $10^6$ CFU/mL in an experimental laboratory setup. A larger application was applied on artificially inoculated (*Pseudomonas fluorescens*~$10^8$ CFU/mL and *Pezicula malicorticis*~$10^6$ CFU/mL) organic apples to test the up-scalability of the PPA treatment. The apples were photographically documented and their texture was analyzed during the 26-day storage phase to investigate the influence of the PPA treatment on the appearance of the apples. The laboratory experiments resulted in a $\log_{10}$-reduction of one to two $\log_{10}$ levels compared to untreated and compressed-air-treated apples. For apples inoculated with *P. fluorescens*, the up-scaled procedure resulted in up to four levels of $\log_{10}$ reduction. In apples inoculated with *P. malicorticis*, the up-scaled procedure resulted in no reduction. This indicates that the application of PPA to organic apples can be effective for bacteria but needs to be optimized for fungi. Therefore, further testing is needed to validate the results.

**Keywords:** food safety; fresh food; microorganisms; microwave discharge; organic foods; *Pezicula malicorticis*; *Pseudomonas fluorescens*; up scalability; warm atmospheric plasma

## 1. Introduction

Apples are an indispensable, valuable food in the human diet [1,2] and deliver high contents of nutrients such as vitamins and oxidants [3]. Recently, the demand for organically produced food has vastly grown [4,5]. To the contrary, a decreasing acceptance for highly processed food or food produced with the use of any pesticides in conventional agriculture is observed [6–14]. Conventional methods for apple sanitation embrace processes such as washing with tap water, cold storage, waxing or hot water dipping [14]. However, these methods are either energy or water consuming, which contradicts a sustainable production method. Novel sanitation methods for food are needed, which helps to meet these growing demands for organic apples. Against that background, organic farming is expected to produce the same number of foods as conventional farming.

This is a goal which is difficult to achieve, and as a consequence, producers try to keep storage losses as small as possible. Throughout the whole production line and in the best case, apples should be stored in a way that does not create unfavorable microflora, which increases mold or spoilage reactions [15]. As shown by various studies, post-harvest-losses yield up to a third of the total harvest, which may be exceeded for organically produced

fruit and vegetables [16–19]. Apples possess a protective cuticle and a natural wax surface, which makes the penetration of microorganisms more difficult. However, the fruit has gas exchange, which is guaranteed by lenticels [20]. Food spoilage is a complex process that can incorporate a variety of native and externally added microorganisms [21]. After damaging the upper layer, pathogenic bacteria like *Pectobacterium carotovorum* can harm all parts of a plant [22–24]. Additionally, studies of Ilyas et al. (2007) showed that the fungi *Penicillium expansum*, *Aspergillus niger*, *A. fumigatus*, *Alternaria tenuis*, *A. tenuissima*, *Cladosporium herbarum*, *Helminthosporium tetramera*, *Mucor racemosus*, *P. italicum*, and *Rhizopus nigricans* were isolated from rotten apples [25]. This variety of apple-damaging microorganisms highlights the desire of many producers to have additional, more potent decontamination methods available. Against that background, non-thermal plasma at atmospheric pressure reveals a way to reduce storage losses [26,27]. Physical plasmas contain excited gas molecules, positively and negatively charged ions, free electrons, reactive oxygen and nitrogen species (RONS), and various free radicals [28–30]. PPA is enriched with chemically reactive compounds such as the reactive nitrogenous and oxygen species (RONS) NO, $NO_2$, and $H_2O_2$ [31–34]. These reactive species can act antimicrobially by damaging DNA, disturbing the microbial signal transduction and impairing the cell envelope [35]. These compounds are thought to inhibit or prevent the growth of native microbial flora on food [36]. Inactivating microorganisms such as molds, bacteria or viruses can preserve apples longer. In general, in the field of biological decontamination, a wide range of plasma applications are currently being researched and applied in pilot projects [26,37,38]. The use of plasma-processed water (PTW) in decontamination processes is a promising way of its application, which showed huge reduction possibilities in terms of water and the use of chemicals [39–41].

The main hypothesis of our work is an anti-microbiological effect of PPA on the microbial community found on apples. Thus, a PPA treatment extends the shelf life of the apples during storage in a cold atmosphere. Since lenticels possibly catch severe contamination, we focused on apple varieties that show well developed lenticels spreading uniformly on the apple surface [42]. The influences of PPA on the native microbial load were checked via a proliferation assay. To the best of our knowledge, the application of PPA and the up-scaled PPA process are novel for the sanitation of apples.

The appearance of the apples was photographically observed, and the crispness of the apples after PPA treatment was detected with a texture analyzer.

## 2. Materials and Methods

### 2.1. Plasma Source

The MidiPLexc was operated at a frequency of 2.45 GHz with a power output of 1.1 kW and a gas flow of 18 standard liter per minute (slm). The gas temperature within the plasma itself reached about 4000 K, which makes this plasma to appear as a quasi-thermal plasma showing properties of both thermal and non-thermal plasmas. Under these parameters, about 3% of the compressed air is processed/functionalized to RONS.

### 2.2. Apple Samples

Organic apples were taken as samples during the experiments. The apples were of the variety Natyra, grown in Germany and bought locally (Greifswald area). Untreated organic apples were taken as references. Samples and references were stored in a refrigerator at 4 °C until the PPA treatment.

### 2.3. Artificial Inoculation of Organic Apples

The organic apples were inoculated artificially with the bacterium *Pseudomonas fluorescens* (DSM 50090/ATCC 13525) and the fungus *Pezicula malicorticis* (DSM 62715) for the Decon-Box experiments (Figure 1). For the inoculation, one colony of the microorganism was taken with an inoculation loop from a 24 h old stock culture and transferred into 100 mL of tryptic soy broth (TSB) (Carl Roth, Karlsruhe, Germany) for *P. fluorescens* or

potato extract glucose broth (PGB) (Carl Roth, Karlsruhe, Germany) for *P. malicorticis*; this procedure was repeated for three separate beakers. These three beakers per microorganism were incubated for 24 h at 70 rpm on an orbital shaker at room temperature. An amount of 400 mL of TSB/PGB was added to all beakers, and all beakers were incubated for another 48 h at 70 rpm a shaker at room temperature. The content of all three beakers was merged into a 3 L beaker. The organic apples were immersed into the suspension for 60 s by a sieve (height $\times$ diameter: 9 cm $\times$ 10 cm). After the inoculation, the apples were stored for 72 h to dry at RT under aseptic conditions before treatment.

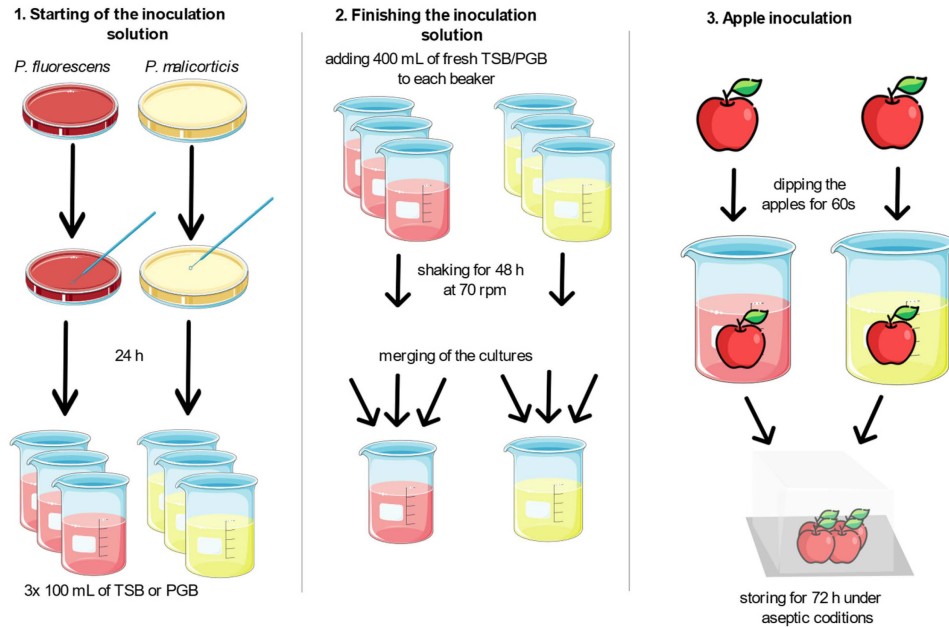

**Figure 1.** Artificial inoculation process of organic apples. The samples, which were investigated during the experiments, were contaminated with *P. fluorescens* and *P. malicorticis*.

*2.4. PPA Treatment of the Native Load on PPA-Treated Apples and Their References*

The apples were put into a tumbler (volume of 0.15 m$^3$, Figure 2) for the PPA treatment.

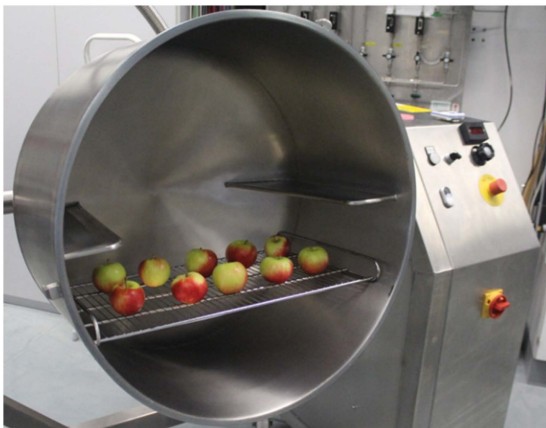

**Figure 2.** A tumbler was used for the PPA treatment of apples. The apples were placed on a grid during every treatment. The tumbler did not tumble as long as apples were loaded.

Therefore, the gas-tight drum of the tumbler was swiveled into a horizontal position. Subsequently, the apples were placed on a stainless-steel grid that crosses the diameter of the drum. The single apples should not touch each other. The lid of the drum houses the inlet wave and the drum were filled from the top with PPA with a gas flowrate of

60 slm and a total volume of 150 standard liter (horizontal inlet). The PPA is heavier than the surrounding air due to $NO_x$ and therefore sinks into the bottom area of the tumbler. Every PPA treatment encompasses a single treatment group for further microbiological investigations, and the total treatment time is 360 s (150 s for filling and 110 s for further reactions in the filled tumbler). The treatment is followed by a 300 s flush with compressed air to remove the PPA from the tumbler. The control for the PPA treatment was a 360 s treatment with compressed air, also followed by a 300 s flush with compressed air.

*2.5. Detection of the Reduction Factors for the Microbial Load*

After the PPA treatment and after selected storage days ranging from directly after the treatment up to 26 days, three apples from the treatment and control were taken from the batch for specific investigations. Therefore, the selected apples were transferred in a closed, sterile plastic box to a microbiological safety cabinet for guaranteeing aseptic conditions. By sterile cotton swaps soaked with TSB, each apple surface was wiped, and subsequently, the swap was transferred into a 15 mL tube with 1 mL of TSB. After a decimal, serial dilution up to 10–8, 10 μL of the diluted suspensions was dropped on each of the different agar plates and spread by tilting the plates. The detection limit was 2 $\log_{10}$-levels (equal to the CFU/mL resulting from a single colony in the lowest dilution). After one day of incubation at RT, the number of colonies (x) of the dilutions was counted manually and finally the colony forming units per milliliter calculated as follows:

$$\frac{\text{CFU}}{\text{mL}} = \frac{\sum_i^n \frac{c_i * f_i}{V}}{n} \text{ for } c_i := \begin{cases} 1 \text{ if } x_i \Lambda_{x_i+1} \neq 0 \\ x_i \text{ if } x_i \neq 0 \\ 0 \text{ if dilution not countable} \end{cases} \quad (1)$$

$x_i$—number of counted colonies;
$f_i$—dilution factor of the i-th dilution;
$V$—volume for the plating in mL;
$n$—number of countable dilutions;

A dilution is declared as "not countable" if the distinction between colonies is no longer possible (overgrown) or if the counted dilution and the next higher dilution are empty. In case of an empty plate (all dilutions have no colonies), the CFU/mL is set to the detection limit.

The antimicrobial properties were described by the calculation of the reduction factor (RF) described in Equation (2).

$$RF = \frac{\sum_{i=1}^n \left( \overline{x_t} - \left[ \log_{10} \left( \frac{\text{CFU}}{\text{mL}} \right) \right]_t \right)}{n} \quad (2)$$

for the average of the $\log_{10}$(CFU/mL) of the reference at day *t* and the decadic logarithm of the CFU/mL of every sample.

*2.6. Up-Scaling of the Labarotory Device to the Pilot-Scaled Decon-Box*

The up scaling of the tumbler-size investigations was carried out as a pilot study in a larger construction named Decon-Box (Figure 3) in the size of 0.2 $m^3$.

The Decon-Box allowed a faster charging with further apple batches, as it is equipped with a transport system with conveyor belts and a treatment chamber open at the top.

The workflow of the Decon-Box is run in four different steps, which are detailed in Figure 4

Step 1 (sample input): A plastic crate loaded with apples was placed on a conveyor belt. The conveyor belt transported the apple crate to the mechanical gripper. The gripper then transferred the crate to the treatment chamber.

Step 2 (PPA treatment): By adding freshly produced PPA, the treatment began. The samples (here apples) could stay here for different treatment times to get an efficient sanitation.

Step 3 (flushing): After the PPA treatment, the mechanical gripper removes the box from the treatment chamber. The box is now transferred to the second separate part (flushing chamber). In the flushing chamber, the remaining PPA is removed by means of compressed air. The compressed air flushing finally terminates the PPA treatment.

Step 4 (sample output): Now the crate is removed from the flushing chamber using the gripper and transferred to the conveyor belt. From this point, the treated and cleaned apples can be transported further (storage or microbial testing).

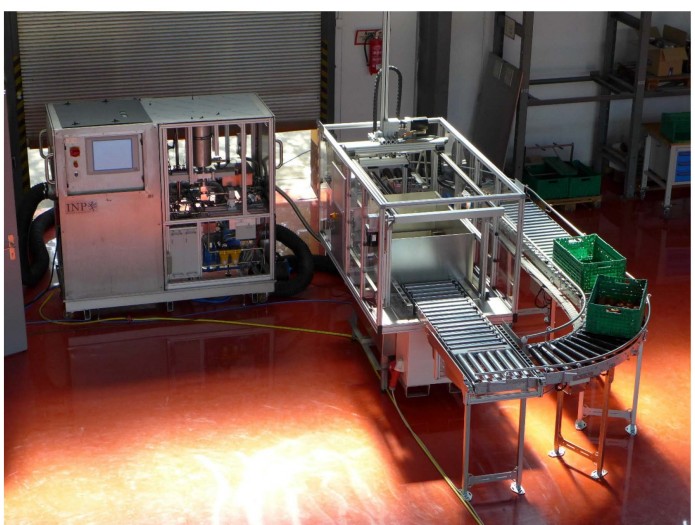

**Figure 3.** Picture of the Decon-Box used for the up-scaled experiments.

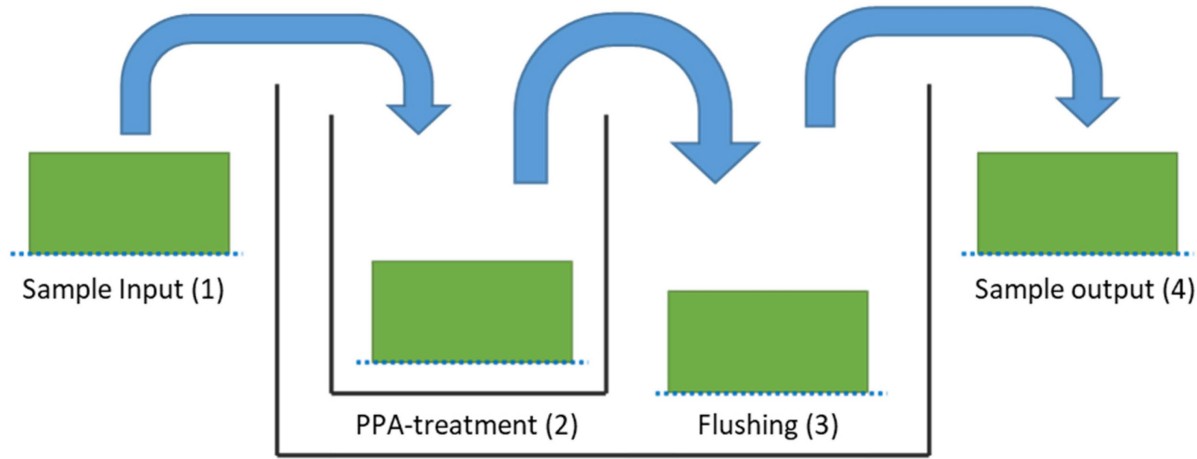

**Figure 4.** Diagram describing the processes in the antimicrobial immersion process in the Decon-Box. The samples, which were investigated during the experiments, were contaminated with *P fluorescens* and *P. malicorticis*. All samples were compared with untreated references.

### 2.7. Data Analysis

Origin 2023b was used to plot the microbial data. Due to the highly overlapping standard deviations (and thus a high probability of overlapping confidence intervals), only descriptive statistics were done. The main reason for the large overlap in standard deviations is biological noise (e.g., differences in native surface load of apples between different apples in a batch). Further experiments are needed to compensate for the biological noise (according to the law of large numbers theorem) and to enable further statistical tools such as outlier reduction without accidental p-hacking. Statistical inference was not performed for the pilot study with the Decon-Box due to lack of sufficient biological and technical replicates.

## 3. Results and Discussion

### 3.1. Decontamination of the Native Load via PPA Treatment

Figure 5a summarizes the time-dependent growth of the culturable part of the native microbiome on apples after PPA treatment. These findings have been compared with two kinds of reference. First, PPA-untreated apples were carried throughout the experiment (negative reference). Additionally, apples treated with compressed air instead of PPA have also been observed (positive reference). For the references, a huge variation of almost half a $\log_{10}$-step is observed. In accordance with our expectations, the number of microorganisms found on the surface increases over the given observation period of 26 days. These findings appeared to be true for both controls. Although not statistically significant, the microbial growth on positive references exceeds that on the negative controls for longer observations times. Nevertheless, the reduction of the microbial load on PPA-treated apples exceeds that on the references. Repeatedly, these offsets are not statistically significant. Based on the presented data, the successes of PPA treatment, i.e., a reduction of the natural microbiome or a hampered growth, cannot be proven based on statistics.

However, Figure 5b summarizes all reduction factors (RF) of all experiments for treated apples. The outcome has been compared with the positive reference as described above (see Section 2.5). Since Figure 5b is only a different presentation when compared with Figure 5a, it just offers the reader a better access to the reduction factors after a PPA treatment. It shows that the bacterial growth is mainly hindered and that a large-scale killing of the organisms is observed. The figure supports the observation that the highest RF are observed for longer treatment times (>14 days, day 3: $1.51 \pm 0.42$ $\log_{10}$-levels vs. day 24: $2.16 \pm 0.00$ $\log_{10}$-levels).

### 3.2. Up-Scaling to the Decon-Box

Essentially, up-scaling was demonstrated by building a larger facility to meet industrial requirements during commercial apple processing. However, the up-scaling step also includes an advanced process management, which offers some advantages in the PPA treatment. Since PPA is heavier than air, the treatment of the large-scale demonstrator was planned as an immersion process. First, of course, the batch size and number per unit time can be increased significantly. Thanks to the open chamber system, several treatment steps can be carried out in parallel. It gives a time-saving component, since the entire process with all sub-steps does not have to be completed first before a new one is started, as is the case in the tumbler. For instance, a batch of apples that has been treated already with PPA can now enter a rinsing step without any delay. In parallel, a subsequent batch is being treated with PPA without further delays. Additionally, the process management also saves energy. In our experimental set-up in the laboratory, the tumbler has to be PPA-evacuated after every treatment. In contrast, the immersive approach used on the demonstrator saves the PPA after every treatment and can be re-used until the active components of the gas are consumed.

The antimicrobial potential of a PPA treatment have been observed for various foodstuff or surfaces [8,43]. However, also plasma-based post-processing of peanuts has been applied on a lab scale and showed promising results [44]. The vast number of studies prove the growing acceptance of plasma-based methods. Nevertheless, although they are not statistically significant and suffering from high error bars, we interpret the results as promising, since the results appeared to be highly reproducible. Antje Fleisch found comparable results [45]. A pilot study with artificially inoculated apples followed as a proof of concept of the Decon-Box (Figure 6).

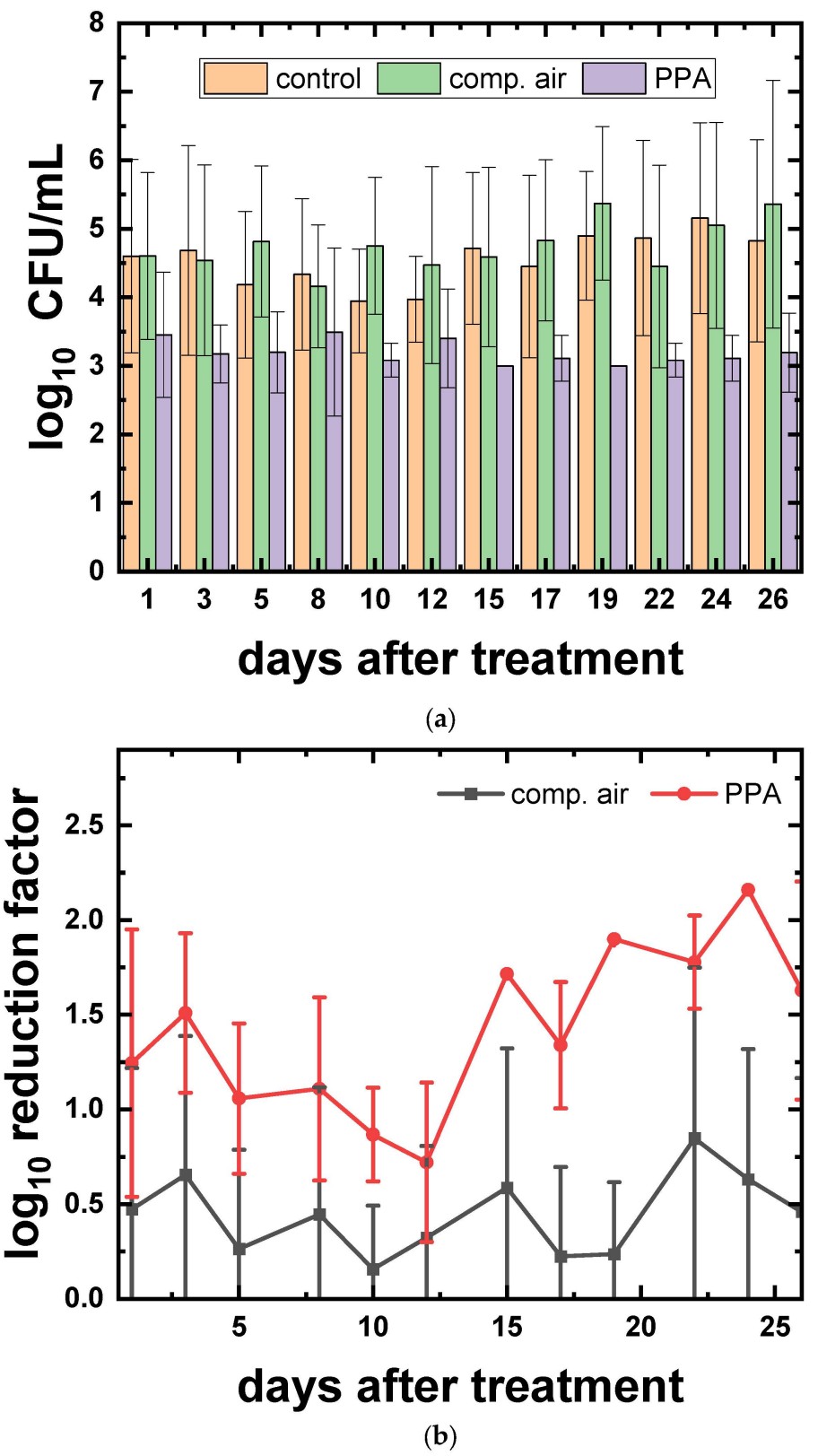

**Figure 5.** (**a**) Results for the native bacterial load on the apple surface investigated under lab-scaled conditions. The picture shows the survivors after a PPA treatment of apples.; (**b**) That picture shows the reduction factors (RF) obtained after a lab-scaled PPA treatment. All experiments with n = 3.

## *P. fluorescens*

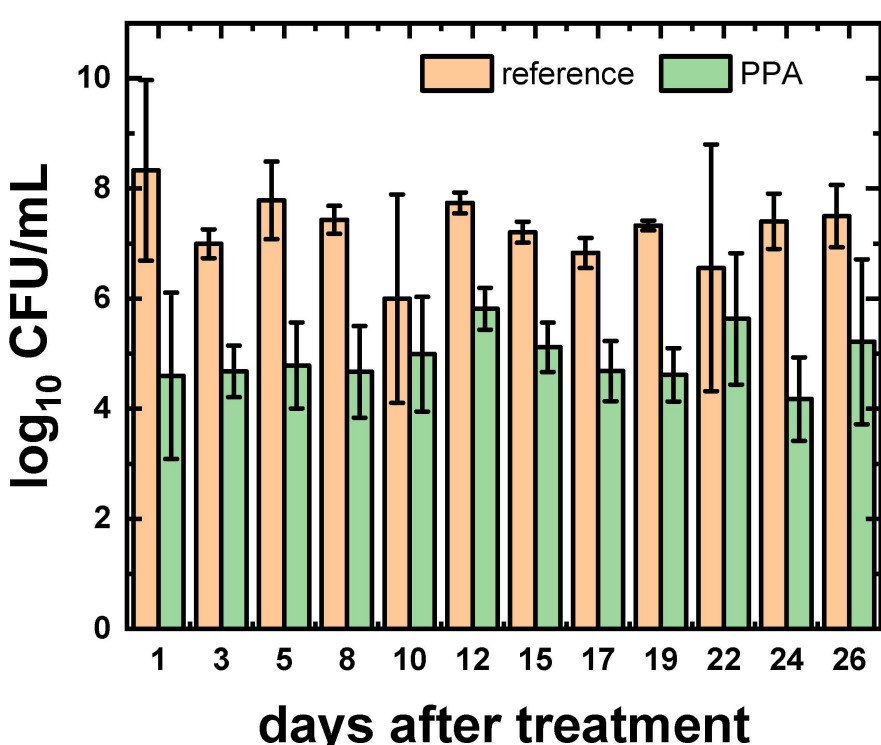

(**a**)

## *P. malicorticis*

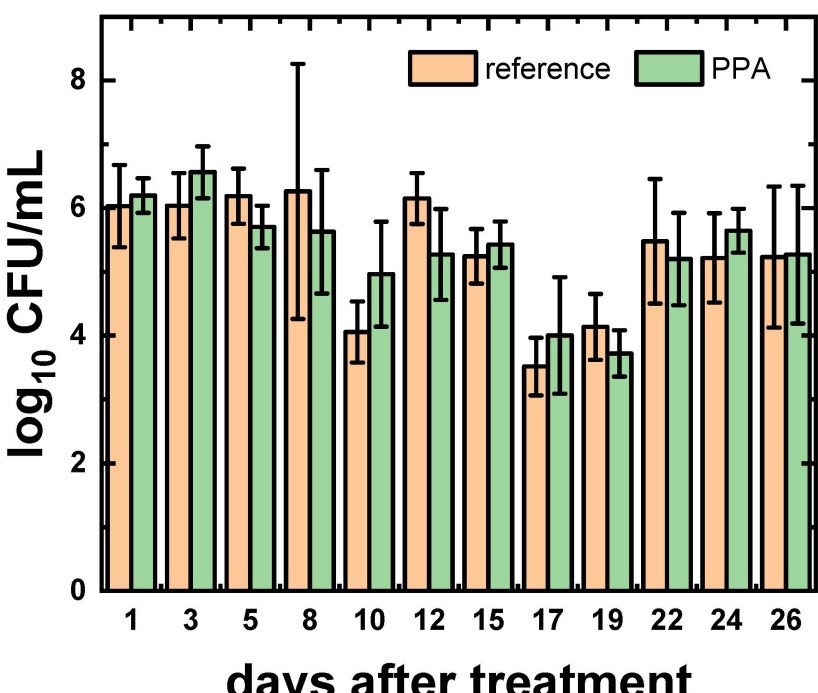

(**b**)

**Figure 6.** *Cont.*

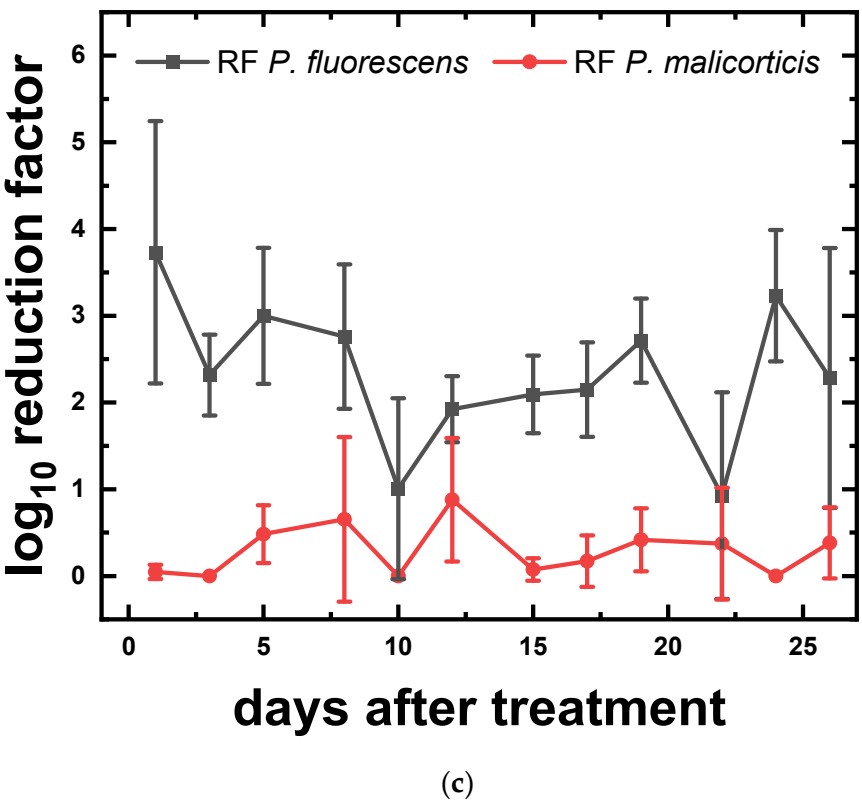

(**c**)

**Figure 6.** (**a**) Survivors of *P. fluorescens* in CFU/mL of PPA-treated apples in up-scaled experiments. The treated apples are compared with untreated references. (**b**) Survivors of *P. malicorticis* in CFU/mL of PPA-treated apples in up-scaled experiments. The treated apples are compared with untreated references. (**c**) Reduction factors (RF) obtained for *P. fluorescens* and *P. malicorticis* inoculated samples in up-scaled experiments. All experiments with n = 2.

Figure 6a shows the time-dependent growth of *P. fluorescens* recovered from the inoculated apples after PPA treatment after selected storage days. These results were compared with an untreated reference. Therefore, untreated apples were carried throughout the experiment and batch-sampled along with the treated apples for the selected storage days. The CFU/mL is lower in the treated apples than in the untreated reference apples.

Figure 6b shows the growth of *P. malicorticis* recovered from the inoculated apples after PPA treatment for selected storage days. These results are compared with those for untreated apples similar to the experiments with *P. fluorescens*. The reference varies less than in Figure 6 (only exception on day 8), but the apples were also less effectively inoculated (~$10^8$ CFU/mL for *P. fluorescens* and ~$10^6$ CFU/mL for *P. malicorticis*). Treatment of the inoculated apples in Figure 6b shows virtually no reduction compared to the untreated reference or compared to the results in Figure 6a.

Figure 6c shows the RF for the treatment of apples inoculated with *P. fluorescens* and *P. malicorticis*, and the difference between the two treatments is even more pronounced. The RF for the *P. fluorescens* treatment ranges from 1 $\log_{10}$-level (day 10) up to ~4 $\log_{10}$-level (day 1). This result shows promising tendencies for the bacterial reduction on the apple surface. The treatment of *P. malicorticis* resulted in no reduction, and therefore further testing on the PPA efficiency on fungi had to be done, since most apple diseases are caused by fungi like the Blue Mold (*Penicillium expansum*).

However, the results in Figure 6a–c cannot be properly tested statistically due to lack of sufficient replicates. Therefore, further large-scale experiments are needed to validate the results. One possible reasoning for the differences between these two treatments could be the differences in the morphological structure of bacteria and fungi. This claim has also to be tested thoroughly in coming experiments.

### 3.3. Quality Measurements of Treated Apples

The quality of produce after a PPA treatment was evaluated based on photographic pictures and texture analysis. Figure 7a summarizes the arithmetic means of all artificially inoculated apples, which underwent PPA treatment. The apples have been compared with their references. Both apples inoculated with *P. fluorescens* and those inoculated with *P. malicorticis* show only a variation in their texture that is within the texture variation of their references. Figure 7b depicts the texture analysis of PPA-treated apples in a pilot scaled experiment. Contrary to the lab-scaled experiments, apples treated in the Decon-Box did not undergo an artificial inoculation step. The apples were treated with different treatment times. Based on the collected data, a 20 min and a 30 min PPA treatment did not reveal any statistical meaningful differences in their crispiness. The matrix of PPA-treated apples stays unchanged throughout the whole set of experiments. These findings support our hypothesis that the antimicrobial active compounds stay solely on the surface of the apples, where they adversely influence the growing microflora. Obviously, these compounds do not enter through the lenticels and do not damage the tissue underneath the apple peel.

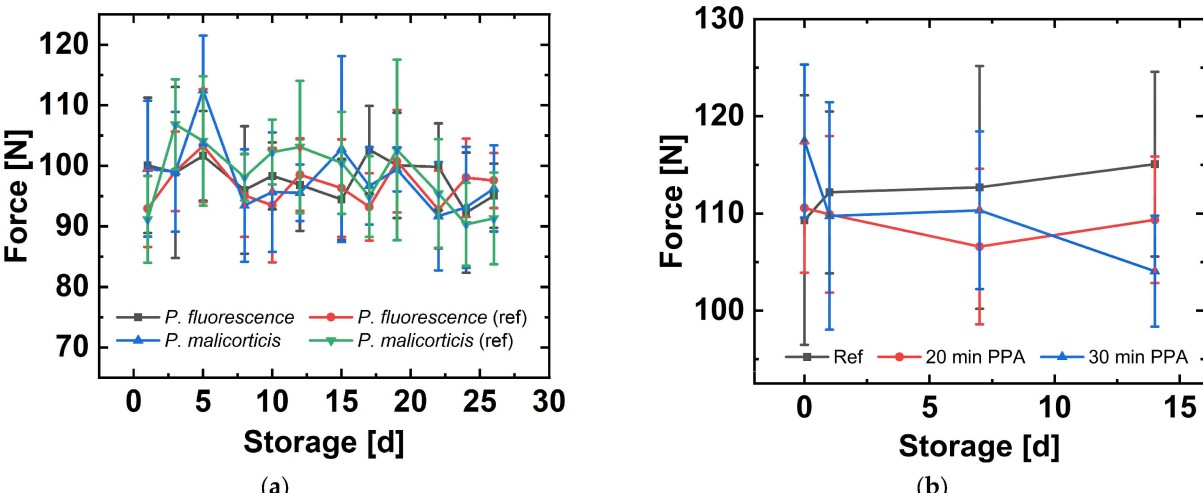

(a) (b)

**Figure 7.** (**a**) Texture measurement of PPA-treated apples on the laboratory scale, which are artificially inoculated with *P. fluorescens* and *P. malicorticis*. The treated apples have been compared with their untreated counterparts (reference); (**b**) texture measurements of PPA-treated apples in the up-scaled experiments, which were conducted in the Decon-Box. Two different PPA treatment times (20 min and 30 min) are compared with their untreated references.

To test the effect of treatment on the storability of organic apples, a control batch and a treated batch of organic apples were observed over a period of 8 weeks (56 days) (Figure 8). As an example, one apple for each of the control and treatment are shown in the following Figure 1 day, 14 days, 7 weeks (49 days), and 8 weeks (56 days) after treatment.

The difference between controls and PPA treatment is marginal and shows that the treatment neither causes visible damage to the apples nor influences the coloration of the apples and does not affect their storability. Therefore, it can be considered that the PPA treatment is an adequate solution to improve the storage of apples, since not only the microbial load is reduced (with the exception of *P. malicorticis* but also the regrowth for *P. fluorescens* and the native microbial inhabitants on the apple surface is impeded. Another indication of the usefulness of PPA treatment is the scalability of the treatment process, taking advantage of the fact that PPA is heavier than air. Thus, the immersion tank only needs to be enlarged and can then be filled with the heavier PPA.

In the case of fruits and vegetables, reactions typically continue to occur even after harvesting [43]. For instance, an apple has to be always considered as a living tissue [44]. Through various different metabolic processes, ripening continues to progress and this contributes to the change in color, in addition to many other phenomena such as the change

in taste, texture, and aroma. This is caused by the degradation of chlorophyll, which provides a green color impression, to various degradation products such as chromoplasts, which produces a color change towards yellow and red tones [12].

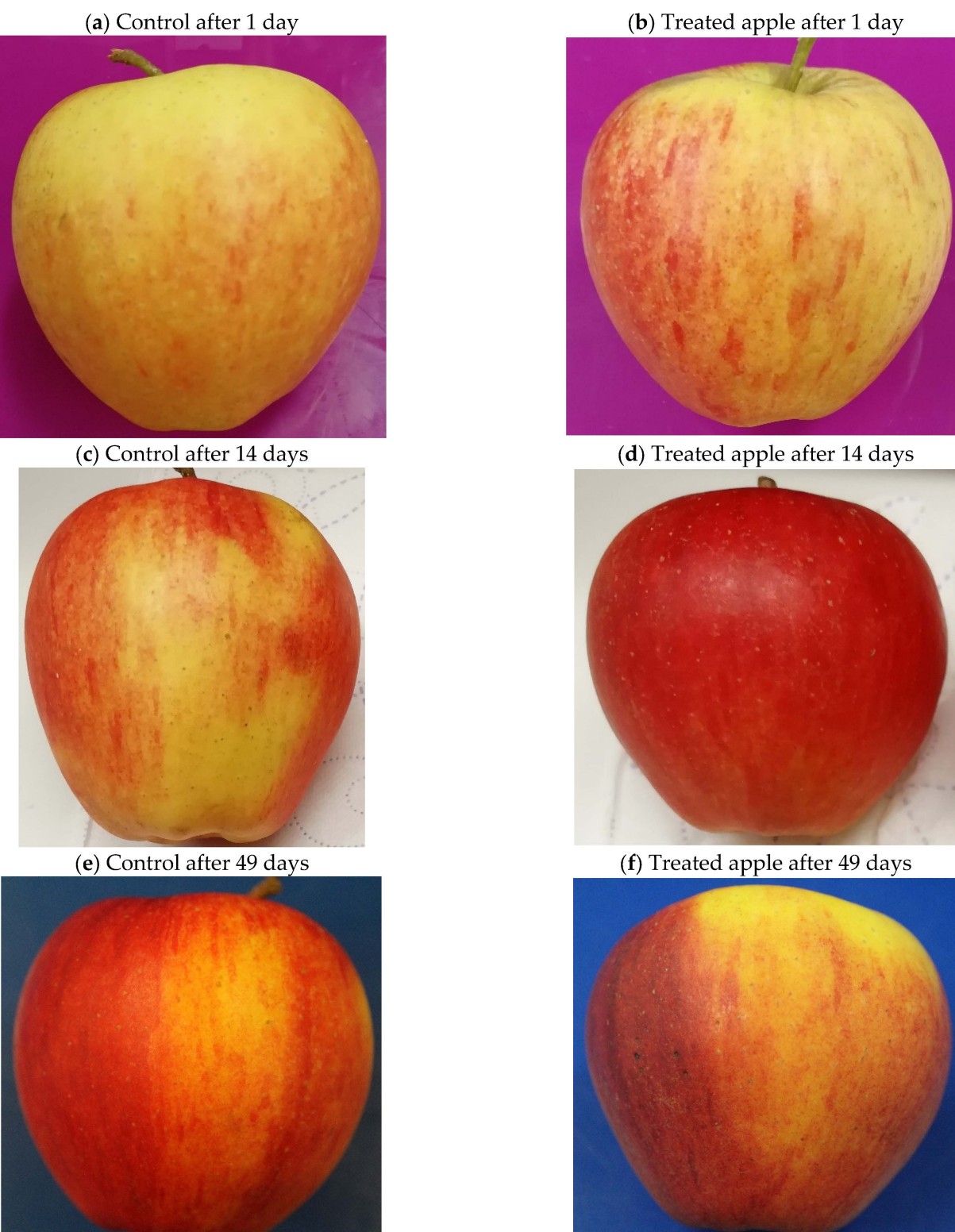

**Figure 8.** *Cont.*

**(g)** Control after 56 days

**(h)** Treated apple after 56 days

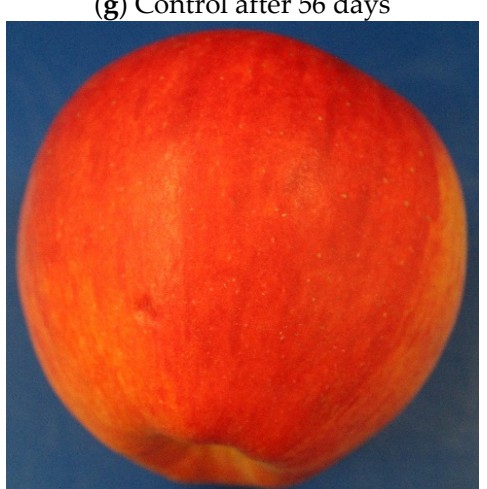

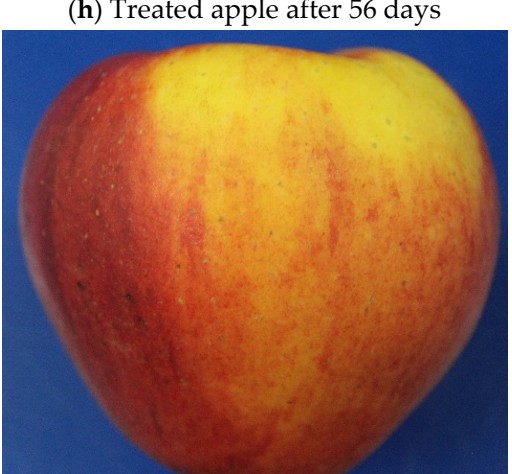

**Figure 8.** Condition of the apples after storage. On the left side are images of control apples (**a,c,e,g**) after 1 day (**a**), 14 days (**c**), 49 days (**e**), and 56 days (**g**) of storage. On the right side are images of PPA- treated apples (**b,d,f,h**) after 1 day (**b**), 14 days (**d**), 49 days (**f**), and 56 days (**h**) of storage.

Regularly, apples undergo a hot-water treatment that improve their storability. This method has been under research for decades [45–47]. Kabelitz (2018) describe a bacteria reduction due to a hot-water treatment up to three $\log_{10}$-steps [12,13]. This magnitude of reduction is also reached for *P. fluoescens* after PPA treatment and a storage period up to 25 d. These findings have been underpinned by quality measurements such as the photographic documentation of the stored apples and the measurement of the apple's texture. However, Herppich et al. (2019) put their apple samples underneath further investigations such as scanning for the vitamin content of the stored apples or chlorophyll fluorescence imaging [48]. Comparable investigations that underpin the promising results of PPA treatment need to be applied, and these studies seem very desirable. The success of PPA treatment in terms of apple quality with sufficient customer acceptance can be ultimately determined.

## 4. Conclusions

Plasma treatment can be a helpful tool to improve the shelf life of apples, but it should not be forgotten that plasma treatment cannot and should not be used alone. Similar to any common hurdle concept in the food sector, the effect of several hurdles should be considered so that after plasma treatment, storage in CA warehouses must still be used. A combined application of PPA and subsequent use of PTW is also possible for synergistic effects in killing germs. Further tests in the future are needed to statistically validate the findings of the given experiments.

**Author Contributions:** Conceptualization, R.W., T.W. and U.S.; methodology, R.W., H.W. and C.W.; validation, R.W., T.W. and H.W.; formal analysis, R.W. and T.W.; investigation, C.W.; resources, U.S.; data curation, R.W. and T.W.; writing—original draft preparation, R.W., T.W. and H.W.; writing—review and editing, T.W. and U.S.; visualization, T.W.; supervision, J.E. and U.S.; project administration, U.S.; funding acquisition, T.W., J.E. and U.S. All authors have read and agreed to the published version of the manuscript.

**Funding:** This research was funded by Federal Republic of Germany, Federal Ministry of Education and Research under the program "PlaVir", grant number 03COV05A; the Federal Ministry for Food and Agriculture (BMEL) under the program "Splash", grant number 2816IP005; the Federal Ministry of Economics and Climate Protection under the funding program "Zentrales Innovationsprogramm Mittelstand (ZIM)", project title "DEADAlus", grant number: 16KN017428.

**Institutional Review Board Statement:** Not applicable.

**Data Availability Statement:** The data presented in this manuscript are available on request from the corresponding authors. The data is not publicly available due to institutional regulations.

**Conflicts of Interest:** The authors declare no conflict of interest regarding the publication of this manuscript.

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
