# Peer review of "Reducing Storage Losses of Organic Apples by Plasma Processed Air (PPA)"

_applsci, doi:10.3390/app132312654_

Round 1

Reviewer 1 Report

Comments and Suggestions for Authors

The presented topic is quite interesting in addressing the problem of organically grown apples. However, the Ms has a number of flaws that needs attention:

1) The abstract does not have any quantitative data. Simple narrations have crowded the text.

2) The technical language has to be improved by a native English speaker

3) statistical data analysis is poorly addressed

4) The presented findings are not well discussed

Comments on the Quality of English Language

The issue of the technical language worrisome. 

Author Response

Dear Reviewer,

thank you for your suggestions. We have tried to incorporate your comments as good as possible.

R: The abstract does not have any quantitative data. Simple narrations have crowded the text
A: Regarding our abstract, we have incorporated your suggestions and implemented quantitative data as well as try to precise more intensively the text.

R: The technical language has to be improved by a native English speaker
A:  We imporved the language according to your suggestions.

R: statistical data analysis is poorly addressed
A: We added information about the data analysis in section 2.7 in the reviewed manuscript beginning at line 203. We did not do any statistical inference due to too high biological noise in the laboratory experiments. Another reason is the lack of sufficient biological and technical replicates in the pilot study, which was caused by the time limitations of the project.

R: The presented findings are not well discussed
A: We restructured the discussion and added more information for the findings

Reviewer 2 Report

Comments and Suggestions for Authors

Reducing storage losses of organic apples by plasma processed air (PPA)

The topic is of interest regarding food safety, and its much more advanced work. The methodology is according to standards. However, the following points should be taken into consideration.

Introduction

Line 50-58: The main fungus for apple contamination is P. expansum, please also include it.

Line 89: delete bracket after degree Celsius.

Figure 1: I think the software used for the preparation of the figure is not licensed, and without a license, the authors cannot use their figures in publication purposes. Please take it into consideration.  

Figure 4 (a) is very compact, we can not observe the parameters, please enlarge it, same Figures 5 (a, b) too

Conclusion

It must be in more detailed form, 

Author Response

Dear Reviewer,

thank you for your suggestions. We have tried to incorporate your comments as good as possible.

R: Line 50-58: The main fungus for apple contamination is P. expansum, please also include it.
A: Penicillium expansum was already included in the original manuscript at line 69, since Penicillium expansum is well known as the main cause of blue mold on apples. To highlight the importance of P. expansum we rearrranged the order.

R: Line 89: delete bracket after degree Celsius.
A: This typo has been overseen and is now correctly deleted.

R: Figure 1: I think the software used for the preparation of the figure is not licensed, and without a license, the authors cannot use their figures in publication purposes. Please take it into consideration.  
A: We changed to a selfmade image based on open source icons.

R: Figure 4 (a) is very compact, we can not observe the parameters, please enlarge it, same Figures 5 (a, b) too
A: We remodeled figure 4 a and b and enlarged figure 5 a,b and c for better visualisation.

Reviewer 3 Report

Comments and Suggestions for Authors

The manuscript titled “Reducing storage losses of organic apples by plasma processed air (PPA)” by Wagner et al. (Ref: Submission ID applsci-2653002), submitted for publication to Applied Sciences is trying to explain the effect of plasma-treated air on reducing losses in organic apples during the storage period. Although this study is interesting, the authors did not do enough research. It was possible to estimate some other biochemical and physiological measurements, for example, the estimation of total soluble solids, the estimation of total soluble sugar, acidity, and the measurement of the enzymatic activity of some enzymes such as pectinase, Phenylalanine ammonia-lyase PAL, and antioxidant enzymes such as catalase, peroxidase, and polyphenol oxidase. Therefore, in my opinion, the research needs further study in order to be publishable in the Journal of Applied Sciences. I recommend major revision and some of these assessments.

Dear Editor

I Hope this mail finds you well.

I have carefully reviewed the manuscript entitled “Reducing storage losses of organic apples by plasma processed air (PPA)” by Wagner et al. (Ref: Submission ID applsci-2653002), submitted for publication to Applied Sciences. As well as, I will answer your concerns about it.

It was possible to estimate some other biochemical and physiological measurements, for example, the estimation of total soluble solids, the estimation of total soluble sugar, acidity, and the measurement of the enzymatic activity of some enzymes such as pectinase, Phenylalanine ammonia lyase PAL, and antioxidant enzymes such as catalase, peroxidase and polyphenol oxidase. Therefore, in my opinion, the research needs further study in order to be publishable in the Journal of Applied Sciences.

Figures are very compact particularly figure 4 (a) and Figures 5 (a, b) too, we cannot observe the parameters.

Author Response

Dear Reviewer,

thank you for your suggestions. We have tried to incorporate your comments as good as possible.

R: It was possible to estimate some other biochemical and physiological measurements, for example, the estimation of total soluble solids, the estimation of total soluble sugar, acidity, and the measurement of the enzymatic activity of some enzymes such as pectinase, Phenylalanine ammonia lyase PAL, and antioxidant enzymes such as catalase, peroxidase and polyphenol oxidase. Therefore, in my opinion, the research needs further study in order to be publishable in the Journal of Applied Sciences.
A: We agree with your suggestion, but we can not deliver the asked experimental data, since our institute is focused on applied plasma physics and therefore we have only limited resources to use and our main focus lies on the antimicrobial efficience. However if we have in future the possibility to do the suggested experiments by equipped partners within the frame of a research project we will be happy to do this and to publish these findings.

R: Figures are very compact particularly figure 4 (a) and Figures 5 (a, b) too, we cannot observe the parameters.
A: We remodeled figure 4 a and b and enlarged figure 5 a,b and c for better visualisation.

Reviewer 4 Report

Comments and Suggestions for Authors

applsci-2653002:

Reducing storage losses of organic apples by plasma processed air (PPA)

L7 → Add correspondence information

Abstract: abstract should state briefly the purpose of the study undertaken, brief mention of experimental aspects (without using abbreviations), highlights of the results and important conclusions based on the obtained results. Therefore, it is suggested that the Abstract should be improved with more data.

Keywords→ arrange in alphabetical order.

The novelty of the work should be emphasized in the introduction section.

Materials and Methods: Add the references of Detection of the reduction factors for the microbial load. Procedures in experimental section must be suitable reference if possible

Add microbial analysis procedure

Materials and Methods: Add the Sources of Materials such as culture medium

Statistical analysis: Which method did you use to compare the significant difference between treatment means? Add Statistical analysis section and analysis data.

In this paper, discussion need to be improved. Results presented need a better discussion. There was no enough discussion or analysis of the results. The author should explain and clearly discuss this part based on scientific knowledge. It is better to compare the results with more similar recent works. And discuss the superiority of the work.

Author Response

Dear Reviewer,

thank you for your suggestions. We have tried to incorporate your comments as good as possible.

R: L7 → Add correspondence information
A: The correspondence information is now added to the manuscript.

R: Abstract: abstract should state briefly the purpose of the study undertaken, brief mention of experimental aspects (without using abbreviations), highlights of the results and important conclusions based on the obtained results. Therefore, it is suggested that the Abstract should be improved with more data.
A:  Regarding our abstract, we have incorporated your suggestions and now mention the respective microorganisms and their basal load in both the laboratory-scaled and pilot experiments.

R: Keywords→ arrange in alphabetical order.
A: We arranged the keywords in alphabetical order.

R: The novelty of the work should be emphasized in the introduction section.
A: We added the information about the novelty in the introduction at line 91 and 92.

R: Materials and Methods: Add the references of Detection of the reduction factors for the microbial load. Procedures in experimental section must be suitable reference if possible
A:  If the reference is needed for the mathematical equation: The equation is a own formal description of the experimental workflow and the workflow is based on "Mikrobiologische Methoden" ISBN: 978-3-8274-1072-6

R: Add microbial analysis procedure
A:  The microbiological analysis is described in section 2.5 of the original manuscript.

R: Materials and Methods: Add the Sources of Materials such as culture medium
A: The sources of the cultural media are added to the manuscript in line 115 and 116.

R: Statistical analysis: Which method did you use to compare the significant difference between treatment means? Add Statistical analysis section and analysis data.
A: We added information about the data analysis in section 2.7 in the reviewed manuscript beginning at line 203. We did not do any statistical inference due to too high biological noise in the laboratory experiments and due to a lack of sufficient biological and technical replicates in the pilot study. Which was caused by the time limitations of project.

R: In this paper, discussion need to be improved. Results presented need a better discussion. There was no enough discussion or analysis of the results. The author should explain and clearly discuss this part based on scientific knowledge. It is better to compare the results with more similar recent works. And discuss the superiority of the work.
A: We restructured the discussion and added more information for the findings. A proper discussion with similar recent works is not suitable, since the application of PPA is noval for apples. A recent publication using PPA is about the decontamination of raw salmon. Comparisons with apples are not applicable due to differences in the surface composition and texture.

Round 2

Reviewer 1 Report

Comments and Suggestions for Authors

The Ms has been improved.

Comments on the Quality of English Language

Acceptable

Author Response

Dear Reviewer #1,

thank you for reviewing our manuscript.

Best regards,

the authors.

Reviewer 3 Report

Comments and Suggestions for Authors

Although the authors improved the manuscript and added some details that help in understanding the study, the research still cannot be published in its current form in the Journal of Applied Sciences because biochemical estimates were not conducted adequately. Therefore, I recommend performing the biochemical assessments previously mentioned in my previous review and resubmitting it again.

Author Response

Dear Reviewer #3

Of course, we totally agree concerning your desire of additional experiments to map the metabolic activity of the treated apples. It is a necessary step to mirror the whole story. However, we are a physical institute, which constructs and promotes plasma sources. Certainly, the current plasma research is moving in a very biological direction, so we also need to answer the more biological questions like the ones raised by you. Unfortunately, we are lacking of equipment and staff for such tests.

But how do we solve our problem of a legitimate demand that we are unable to meet?

In my opinion we only have two possibilities:

  1. You reject our manuscript due to the unsatisfiability of your demand.
  2. The editor has to decide whether the paper can be published or not.

Nevertheless, we thank you very much for your inspiring and challenging suggestions. And we hope to meet your demands in our future publications.

Thank you very much for your patience!

The authors.

Reviewer 4 Report

Comments and Suggestions for Authors

No Comment

Author Response

Dear Reviewer #4,

thank you for reviewing our manuscript.

Best regards,

the authors.